# Asymmetry in Atypical Parkinsonian Syndromes—A Review

**DOI:** 10.3390/jcm13195798

**Published:** 2024-09-28

**Authors:** Patryk Chunowski, Natalia Madetko-Alster, Piotr Alster

**Affiliations:** Department of Neurology, Medical University of Warsaw, 03-242 Warsaw, Poland; natalia.madetko@wum.edu.pl (N.M.-A.); piotr.alster@wum.edu.pl (P.A.)

**Keywords:** atypical parkinsonism, PSP, symmetry, neurodegeneration, parkinsonism, movement disorders

## Abstract

**Background/Objectives:** Atypical parkinsonian syndromes (APSs) are a group of neurodegenerative disorders that differ from idiopathic Parkinson’s disease (IPD) in their clinical presentation, underlying pathology, and response to treatment. APSs include conditions such as multiple system atrophy (MSA), progressive supranuclear palsy (PSP), corticobasal syndrome (CBS), and dementia with Lewy bodies (DLB). These disorders are characterized by a combination of parkinsonian features and additional symptoms, such as autonomic dysfunction, supranuclear gaze palsy, and asymmetric motor symptoms. Many hypotheses attempt to explain the causes of neurodegeneration in APSs, including interactions between environmental toxins, tau or α-synuclein pathology, oxidative stress, microglial activation, and vascular factors. While extensive research has been conducted on APSs, there is a limited understanding of the symmetry in these diseases, particularly in MSA. Neuroimaging studies have revealed metabolic, structural, and functional abnormalities that contribute to the asymmetry in APSs. The asymmetry in CBS is possibly caused by a variable reduction in striatal D2 receptor binding, as demonstrated in single-photon emission computed tomography (SPECT) examinations, which may explain the disease’s asymmetric manifestation and poor response to dopaminergic therapy. In PSP, clinical dysfunction correlates with white matter tract degeneration in the superior cerebellar peduncles and corpus callosum. MSA often involves atrophy in the pons, putamen, and cerebellum, with clinical symmetry potentially depending on the symmetry of the atrophy. The aim of this review is to present the study findings on potential symmetry as a tool for determining potential neuropsychological disturbances and properly diagnosing APSs to lessen the misdiagnosis rate. **Methods:** A comprehensive review of the academic literature was conducted using the medical literature available in PubMed. Appropriate studies were evaluated and examined based on patient characteristics and clinical and imaging examination outcomes in the context of potential asymmetry. **Results:** Among over 1000 patients whose data were collected, PSP-RS was symmetrical in approximately 84% ± 3% of cases, with S-CBD showing similar results. PSP-P was symmetrical in about 53–55% of cases, while PSP-CBS was symmetrical in fewer than half of the cases. MSA-C was symmetrical in around 40% of cases. It appears that MSA-P exhibits symmetry in about 15–35% of cases. CBS, according to the criteria, is a disease with an asymmetrical clinical presentation in 90–99% of cases. Similar results were obtained via imaging methods, but transcranial sonography produced different results. **Conclusions:** Determining neurodegeneration symmetry may help identify functional deficits and improve diagnostic accuracy. Patients with significant asymmetry in neurodegeneration may exhibit different neuropsychological symptoms based on their individual brain lateralization, impacting their cognitive functioning and quality of life.

## 1. Introduction

Atypical parkinsonian syndromes (APSs) are a group of neurodegenerative diseases featuring the intracellular accumulation of amyloidogenic proteins, such as α-synuclein or tau, based on diverse pathologies [1]. Among the diseases in this group are dementia with Lewy bodies (DLB), multiple system atrophy (MSA), progressive supranuclear palsy (PSP), and corticobasal degeneration (CBS). PSP and CBS are rare disorders; more precisely, the prevalence rates of MSA, PSP, and CBS generally fall below 10 per 10^5^ individuals [2], with PSP affecting approximately 3 to 7 individuals per 100,000, CBS having a prevalence of around 5 to 7 cases per 100,000, and MSA having a prevalence ranging from 1.9 to 6.2 per 100,000 [2,3]. APSs are commonly primarily misdiagnosed as Parkinson’s disease (PD) due to their symptoms frequently resembling those typical of PD in the initial stages [4]. Common symptoms of PD and APSs are collectively referred to as parkinsonism and include rest tremor, rigidity, bradykinesia, and postural instability [5], which is caused by a decline in muscle strength. A significant number of these symptoms can be traced back to compromised skeletal muscle health [6]. Symptoms typical of APSs that differentiate these two conditions are, among others, early autonomic dysfunction, early falls, early cognitive impairment, early bulbar dysfunction, and poor or absent response to levodopa [7,8,9,10]. A positive response to levodopa treatment was confirmed in 44% of PSP cases, 52% of MSA patients, and up to 12.5% of individuals with CBS in comparison to PD subjects, where an efficient levodopa response was observed in 91% of cases [11]. Clinically, patients with MSA present with a variable combination of parkinsonian, cerebellar, autonomic, and pyramidal signs. PSP, on the other hand, is characterized by parkinsonism, supranuclear gaze palsy, postural instability, and early falls, which can appear in different subtypes. In contrast, the typical clinical presentation of CBD is corticobasal syndrome (CBS), characterized by parkinsonism, dystonia, myoclonus, cortical sensory loss, ideomotor apraxia, alien limb phenomena with a predominantly asymmetrical distribution, and additional cognitive and behavioral impairments [12]. However, these clinical signs do not offer complete certainty in establishing an APS diagnosis, and many of the symptoms only become apparent in the later stages of the disease [13]; because of this, inadequate diagnosis can occur in up to 24% of PSP cases [14]. It is estimated that individuals meeting MSA criteria were diagnosed properly in 62% of samples [15]. Among patients who received a clinical diagnosis of CBS, only 50% had a confirmed diagnosis of CBD [16]. These data indicate the need to find an additional diagnostic tool. Symmetry is cited as one of the main diagnostic criteria in the context of CBS [8] but only an additional criterion in PSP [9]. According to the diagnostic criteria, it is assumed that PSP is the most symmetrical disease, and on the other end of the spectrum is CBS [8,9]. The diagnosis of Parkinson’s disease is determined by the occurrence of unilateral symptoms, especially at the beginning of the disease [17], and these symptoms are consistent with APSs; therefore, early-stage symmetry can potentially indicate a particular APS subtype, which should prompt further diagnostic evaluation, even at an early stage of the disorder. This is why potential symmetry evaluation seems to be a promising additional diagnostic method for differentiating between APSs and PD. However, the current criteria do not address MSA or particular subtypes of tauopathic illnesses. The aim of this review is to present the study findings on the potential symmetry evaluation as a tool for making proper diagnoses and identifying potential cognitive and emotional dysregulation, such as depression, anxiety, agitation, appetite changes, or increased levels of aggression in APSs.

## 2. Materials and Methods

A review was conducted in order to select suitable studies evaluating subjects with APSs, especially regarding symmetry, published between 1993 and 2024. The search algorithm used the following search terms in Medical Literature, Analysis, and Retrieval System Online (MEDLINE) and the Cochrane Central Register of Controlled Trials (CENTRAL): “asymmetry in atypical parkinsonian syndromes”, “MRI in atypical parkinsonian syndromes”, “PET in atypical parkinsonian syndromes”, “SPECT in atypical parkinsonian syndromes”, “transcranial sonography in atypical parkinsonian syndromes”, “progressive supranuclear palsy”, “corticobasal syndrome”, “corticobasal degeneration”, “multisystem atrophy”, “asymmetry in progressive supranuclear palsy”, “asymmetry in multisystem atrophy”, “asymmetry in corticobasal syndrome”, “asymmetry in corticobasal degeneration”. The review was limited to studies enrolling at least 3 subjects and to articles published in Polish or English. Studies conducted on fewer than 3 participants, those with duplicate cohorts, and those in languages other than Polish or English were excluded from this review.

## 3. Atypical Parkinsonism’s Clinical Symptom Asymmetry

Differentiating between PD and APSs can be challenging due to overlapping clinical manifestations, especially in the initial stages [18]. Progressive supranuclear palsy-Richardson syndrome (PSP-RS) is considered the most common and symmetrical subtype of PSP. Using the Neuroprotection and Natural History in Parkinson Plus Syndromes (NNIPPS) criteria [19], 18 patients were classified as having PSP, and 7 of them (38.9%) were further classified into the PSP-RS group [20]. Another study, in which 50 PSP patients were thoroughly examined, concluded that 28 of them (56%) had PSP-RS [21]. In this study, 33 patients were excluded due to incomplete data (n = 7) or because they qualified for other phenotypes, such as probable or possible PSP-RS and progressive supranuclear palsy-parkinsonism (PSP-P) or possible progressive supranuclear palsy–corticobasal syndrome (PSP-CBS) (n = 26). Among 334 patients diagnosed with PSP, the majority (72%) were identified as having the PSP-RS subtype using the Movement Disorder Society (MDS) criteria for PSP from 2017 [22]. In total, over 400 people were examined, revealing a significant numerical predominance of PSP-RS diagnoses over other types of PSP. Motor characteristics such as bradykinesia, rigidity, rest tremor, and dystonia were assessed using specific elements from the NNIPPS scale. Features that showed variations in scoring between the right and left sides were considered asymmetric. PSP patients might display asymmetric limb bradykinesia and rigidity [20]. The research demonstrated that parkinsonian syndrome symptoms, including bradykinesia, rigidity with tremor, and other symptoms, such as dystonia or myoclonus, exhibited asymmetry in a significant proportion of PSP-RS cases (53.6%, 21.4%, and 17.9%, respectively). Additionally, a smaller percentage of patients displayed asymmetry in higher cortical functions, such as limb apraxia (35.7%). The higher cortical function assessment comprised 24 activities split evenly, with 12 focusing on symbolic gestures and 12 on nonsymbolic gestures. In certain instances, PSP may manifest symptoms associated with CBS, such as apraxia, the alien limb phenomenon, and the loss of cortical sensory functions. CBS is often suspected when there are asymmetrical signs and symptoms during an individual’s lifetime [23]. Both PSP-P and PSP-CBS are considered to express notable asymmetric clinical features. Regarding motor symptoms, rigidity and bradykinesia exhibited asymmetry in over half of 50 participants with different PSP types, according to the MDS criteria, whereas tremor was asymmetric in only 14.3% of cases of probable PSP-RS [22]. In evaluating the eyes, up- and downward asymmetry in vertical saccade velocity was observed in approximately 34% of 80 patients suffering from PSP [24]. The symmetry in eye movement disorders is most likely correlated with the symmetry of midbrain atrophy. In 18 individuals in a small cohort study (PSP-RS (n = 7), PSP-P (n = 3)), progressive supranuclear palsy-pure akinesia with gait freezing (PSP-PAGF) (n = 2), progressive supranuclear palsy-frontotemporal dementia (PSP-FTD) (n = 4), progressive supranuclear palsy-apraxia of speech syndrome (PSP-AOS), and progressive non-fluent aphasia (PSP-PNFA) (n = 2) exhibited symmetrical symptoms [20]. In a retrospective study that examined 25 patients, 9 were classified as having probable PSP-RS, while 16 were assigned to the possible PSP-CBS group. Both groups exhibited asymmetrical dystonia of the limbs, but they did not present cortical signs. However, it should be noted that focal limb dystonia might be an early feature of CBS when compared to cortical dysfunction. Early asymmetric limb dystonia might indicate evolving PSP-CBS rather than PSP-RS and thus requires longitudinal patient follow-up. However, the MDS-PSP criteria do not consider asymmetric dystonia when classifying potential PSP-CBS [25]. Symmetry in PSP can vary, leading to the identification of two subgroups that highlight the degree of symptom symmetry. A hemi-PSP classification was assigned to PSP patients displaying notable asymmetry in rigidity, dystonia, or bradykinesia, in contrast to those with a symmetric presentation (symPSP). On the other hand, the least symmetrical representative of APSs is CBS, which involves the presence of apraxia, cortical sensory loss, and/or alien limb phenomena, along with an asymmetric hypokinetic disorder, often accompanied by limb dystonia. According to the diagnostic criteria consensus, 23 individuals were diagnosed with PSP, and 8 patients were diagnosed with CBS. Of those with PSP, 14 presented with a symmetric illness manifestation (symPSP, 60.9%), whereas 9 showed a markedly lateralized manifestation (hemi-PSP, 39.1%). In all PSP-P patients, asymmetric symptoms were observed, two out of two PAGF patients had symmetric symptoms, and all CBD patients expressed symptom asymmetry [26], which is consistent with the CBS diagnostic criteria. PSP might also resemble MSA, as mentioned above for PSP-CBS, especially when they are characterized by early falls and a supranuclear palsy affecting vertical gaze [27]. CBS might mimic PSP in terms of symmetry, especially when CBS has a genetic basis. Based on the symmetry of symptoms, 33 CBS patients were divided into two subgroups: S-CBD (n = 5) and CBS (n = 28). The study compared clinical symptoms and signs between the two groups, regardless of whether the manifestation was symmetric or asymmetric. When categorizing the signs as motor, behavioral, language, cognitive, apraxia, and sleep issues, key differences emerged. Notably, behavioral changes were significant in symmetric CBD but not present in CBS. Conversely, language issues, limb apraxia, and several motor symptoms, such as axial rigidity, bradykinesia, myoclonus, alien limb, and the Babinski sign, were prevalent in CBS but absent in cases of symmetric CBD. Both groups exhibited motor symptoms, like limb rigidity, falls, parkinsonism, rest tremor, gaze palsy, hyper-reflexia, and dysphagia, as well as cognitive symptoms, including memory loss and difficulty with calculations. When motor symptoms appeared in CBD and they manifested symmetrically. This disease is conventionally viewed as a sporadic disorder; however, a positive family history of neurodegenerative disease was more prevalent in cases of symmetric corticobasal degeneration (S-CBD), implying a potential genetic predisposition to the development of symmetric degeneration [23]. This is why distinguishing S-CBD from PSP-RS is very hard. Five patients were reported to initially present with highly asymmetric parkinsonism accompanied by dystonia, initially diagnosed as CBS. However, as the disease progressed (approximately five years after disease onset), the parkinsonism became less asymmetric, and the patients developed autonomic features and respiratory issues, ultimately leading to a revised diagnosis of multiple system atrophy-parkinsonism (MSA-P). In this condition type, clearly distinguishable asymmetry is generally rare; because of asymmetric dystonia of the limbs and myoclonic jerks, individuals were incorrectly diagnosed with CBS [28]. According to Kouri et al.’s study, patients with corticobasal degeneration-corticobasal syndrome (CBD-CBS) commonly show asymmetric limb stiffness (asymmetric in 100%), apraxia (asymmetric in 91%), localized limb dystonia, myoclonus, and cortical sensory impairments, more so than those with corticobasal degeneration-Richardson syndrome (CBD-RS). Rigidity is present in 20% of CBD-RS cases, but in PSP-RS, all cases are symmetric. While limb apraxia is infrequent in both CBD-RS and PSP-RS, it appears symmetrically when it occurs, in contrast to the pronounced asymmetric limb apraxia in CBD-CBS patients. Signs of both pyramidal and extrapyramidal disorders are observed at comparable rates among these groups [29]. This may indicate that the degree of symmetry is not directly proportional to the extent of neurodegeneration. MSA patients, in most cases (but not always), have symmetrical rigidity and bradykinesia [30]. Of 16 MSA patients, only 7 showed symmetrical disease symptoms, such as tremor or rigidity [31]. In another study, 20/23 (87%) MSA-P patients and 8/12 (67%) MSA-C individuals exhibited clinical symptom asymmetry [32], which is consistent with Van Laere and colleagues’ research, in which a significant difference in the degree of clinical lateralization between PSP and MSA was observed [33]. The symmetry rate is summarised in the Figure 1 and Figure 2.

## 4. Brain Lateralization

Left-handed individuals make up approximately 10% of the population, a rarity that is believed to result from the development of language lateralization in the left hemisphere, which has led to a predominantly right-handed population [34]. The relationship between language lateralization and handedness has been extensively studied. Furthermore, the differences in the brains of left- and right-handed individuals extend beyond language lateralization, encompassing variations in motor and somatosensory networks [35]. There is a belief that hand dominance and skillful learned control are intricately linked. The development and activation of the primary motor cortex show mirror-opposite patterns in right- and left-handers. An analogous shift in dominance to the right hemisphere for different types of praxis might be expected in left-handed persons [36]. Neuroimaging studies have increasingly highlighted these differences through functional activation analyses [35].

## 5. Imaging Tests

The current diagnostic criteria for PSP and MSA consider imaging of atrophy or hypometabolism using magnetic resonance imaging or PET. CBS criteria do not take into account magnetic resonance imaging (MRI) and positron emission tomography (PET) results [7,8,9].

## 6. MRI

Patients with PSP-RS typically present with significant midbrain atrophy. The characteristic imaging features include the “hummingbird” sign, alternatively termed the “penguin” sign, which describes the flat or concave appearance of the midbrain. Additionally, the “morning glory” sign corresponds to the concave aspect of the lateral margin of the midbrain tegmentum on axial slices, and this feature is also observed in PSP-RS. While these imaging characteristics are indicative of PSP-RS, with a specificity of 99.5% for the “hummingbird sign” and 97% for the “morning glory sign”, both of them exhibit low sensitivity [4,37,38,39]. In PSP patients, decreased midbrain [40,41], total brain, and thalamus volumes, along with an increased volume of the ventricles, were observed at both 6 and 12 months when compared to the initial assessment. However, the difference in midbrain volume turned out to be the largest, so the speed of midbrain shrinkage seems to be the most effective measure of PSP progression. It turned out to be a better indicator than the pons-to-midbrain ratio [41]. In each of the mentioned studies, MRI changes did not show clinically significant asymmetry, which is consistent with the fact that PSP-RS is considered the most symmetrical form among atypical parkinsonism syndromes. The volumes of the putamen and globus pallidus within the basal ganglia are observed to be symmetrically smaller compared to those in PD. Additionally, the volume of the thalamus is also symmetrically smaller in PSP-RS. Frontal brain atrophy is observed as well [4]. Atrophy also occurs in cases of CBS. The posterior frontal and parietal lobes consistently exhibited more severe atrophy than other lobes, while atrophy in the occipital lobe was rarely observed. Atrophy in the temporal and anterobasal frontal lobes was frequent but less severe than in the posterior frontal and parietal lobes. Cerebral peduncle atrophy was noted in seven patients, with six of them exhibiting atrophy ipsilateral to the dominant atrophic cerebral hemisphere. Additionally, the dynamics of the atrophy were significant over the course of one year in the patient cohort. Initially, 18 patients presented with asymmetrical parietal cortical atrophy contralateral to the more affected limbs, which increased to 23 patients at follow-up. Initially, 8 patients exhibited both asymmetrical frontal and parietal atrophy, predominantly affecting the posterior frontal areas contralateral to the more affected side, which rose to 12 patients at follow-up. Bilateral parietal atrophy was initially seen in 5 patients, which increased to 10 patients. Subcortical atrophy of the white matter in the corresponding parietal area, initially observed in 7 patients with concurrent lateral ventricle enlargement, was later identified in 15 patients. Lastly, the loss of the putamen signal on high-field T2-weighted MRI was noted in 13 patients initially, which increased to 17 patients after one year. These findings highlight progressive brain atrophy, particularly in the parietal and frontal regions, and increasing signal loss in the putamen over time. It is postulated that CBS subjects exhibit increased levels of depression, anxiety, agitation and aggression, and changes in appetite compared to those with PD, and more than 87% of individuals suffering from CBS developed apathy (in the case of PD, it was 24%) [42], which might be connected with the above-mentioned parietal atrophy, especially when the non-dominant cerebral hemisphere is affected.

Apathy and depression were observed in 13 out of 26 MSA-P patients. Compared to the controls, these patients demonstrated significant cortical thinning in the fronto-temporal-parietal regions, as well as atrophy in the periaqueductal gray matter, left cerebellar hemisphere, left pallidum, and bilateral putamen [43].

Diffusion tensor imaging (DTI) and DTI-based fiber tractography are now commonly used techniques for assessing and illustrating the path, position, and size of significant white matter pathways, including the arcuate fasciculus, corticospinal tract, and optic radiation [44]. In one study, 27 individuals—9 with CBS and 18 with definite or probable PSP (half of them pathologically confirmed)—were prospectively recruited and underwent 3.0 T DTI. The most prominent areas displaying decreased fractional anisotropy (FA) and heightened mean diffusivity (MD) in subjects with CBS were identified in supratentorial regions, including the body of the white matter regions of the premotor, prefrontal, and motor cortices, as well as the middle cingulate cortex and corpus callosum, in comparison to the control group. The observed abnormalities in both FA and MD exhibited asymmetry, with a more pronounced impact on the hemisphere that was more affected. Additional regions showing reduced FA and increased MD included the parietal lobes and fornix, along with the splenium of the corpus callosum in the more affected hemisphere. Furthermore, diminished FA was noted in the pons and cerebellum in both the more and less affected hemispheres, as well as in the superior cerebellar peduncle of the less affected hemisphere. Elevated MD was also evident in the thalamus and posterior temporal white matter in the more affected hemisphere. In contrast to CBS, the most notable areas displaying diminished FA and heightened MD in individuals with PSP were identified in infratentorial brain regions, including both the bilateral superior cerebellar peduncles and midbrain, as compared to the control group. Additionally, reduced FA was observed bilaterally in the body of the corpus callosum, middle cingulate bundle, pons, fornix, and white matter of the premotor and prefrontal cortices [45]. DTI research on patients with PSP-RS shows higher MD in the putamen, caudate nucleus, and globus pallidus when compared to individuals with PD and healthy participants. On the other hand, in CBS, specific degeneration patterns above the tentorium cerebelli, characterized by asymmetric shrinkage in areas like the superior parietal lobe, posterior frontal lobe, and basal ganglia, can be observed. Atrophy may be asymmetrical, but the absence of pronounced asymmetry does not exclude the possibility of CBS [46]. Additionally, there is deterioration in the splenium and the body of the corpus callosum, the middle cingulate bundle, and the white matter tracts in the superior parietal, premotor, and motor regions [47]. On the other hand, a functional MRI study focuses more on task performance than simple MRI. A sample of CBS patients with limb apraxia showed diminished activity in the premotor cortex alongside heightened activity in the parietal cortex, suggesting a potential compensatory mechanism through neural recruitment [48]. Planning non-functional or structural grasp-to-pass movements for inconveniently oriented tools, regardless of which hand was used, significantly activated the left parietal and prefrontal nodes more than simple, undemanding functional grasps [49]. Therefore, because of the compensation between the premotor and parietal cortices, apraxia is a less expressed symptom. Key MRI findings in MSA include atrophy of the cerebellum and brainstem. The “hot cross bun” (HCB) sign, characterized by cruciform T2 hyperintensity in the pons, is a hallmark of MSA, and it has sometimes been noted in other neurological conditions, such as autopsy-confirmed CBD [50] or autoimmune cerebellar ataxia [51]. These features are usually symmetrical [52]. Coronal fluid-attenuated inversion recovery (FLAIR) images confirmed asymmetric atrophy in the posterior putamen, resulting in the near-total loss of neurons in this structure. This MRI finding is often called the “putamina rim” sign. The putamen and the globus pallidus are parts of the striatum. The striatum plays a crucial role in numerous brain functions, such as motor control and learning, language processing, reward mechanisms, and cognitive functioning [53]. In the neuropsychological context, Broca’s and Wernicke’s areas (located in the frontal and temporal lobes, respectively) are involved in speech functions, but the role of the putamen in language involves a network of coactivations in both the left and right putamina, with the left putamen playing a significant role in additional language functions, including bilingual language processing [54]. Hypointensity in the atrophic putamen on coronal FLAIR images suggested iron deposition linked to neurodegeneration, distinguishing it from age-related changes in the external globus pallidus. The patient’s initial asymmetric parkinsonism aligns with previous reports of asymmetrical parkinsonism in early-stage MSA-P cases [55].

## 7. Single-Photon Emission Computed Tomography (SPECT)

A DaTscan examination is used for dopamine transporter density imaging with Ioflupane I123, which is a good marker of presynaptic nigrostriatal dysfunction [56]. The use of presynaptic dopaminergic imaging, such as dopamine transporter imaging, is constrained in distinguishing among different parkinsonian syndromes. This limitation results from the fact that the presynaptic dopaminergic system is affected not only in APSs but also in PD [57]. However, there is a more precise and meticulous imaging method based on DaTscan.

Dopamine transporter single-photon emission computed tomography (DaT-SPECT) software facilitates the automated computation of three quantitative indices: the specific binding ratio, putamen-to-caudate ratio, and asymmetry index (AI), which are quantitative parkinsonian type indices calculated from DaT-SPECT. The AI is often used to indicate the asymmetry of reduced striatal 123I-Ioflupane accumulation on DaT-SPECT. AI values on DaT-SPECT measured in the PD group (n = 311), PSP group (n = 33), and MSA-P group (n = 20) were significantly greater than those observed in the control group (n = 137) [58]. The asymmetry of striatal 123I-Ioflupane binding appears to be less prominent in MSA and PSP (n = 24 cumulatively) compared to PD (n = 48), although no statistical difference was observed. Greater binding symmetry was detected in MSA-P compared to PD, and this was also observed in another study. It remains unclear how the asymmetry of motor symptoms at various stages of PD correlates with DaT-SPECT imaging and whether the asymmetric reduction in striatal 123I-Ioflupane accumulation on DaT-SPECT will prove valuable in distinguishing PD from PSP and MSA-P [59,60]. In patients with MSA, a decrease in perfusion was observed bilaterally in the cerebellar cortex and vermis, along with reduced perfusion in the left and right posterior putamina. The inverse contrast indicated higher relative perfusion in the occipital and right primary motor cortices compared to IPD [35]. In another study, 16 CBS cases were examined. All of these patients had cerebral atrophy, which was symmetric in 81%. Cerebral peduncle atrophy stood out as the most prominent imaging feature of CBS. Atrophy in the midbrain tegmentum was observed in eight patients, with three of them presenting vertical gaze palsy. Additionally, atrophy in the corpus callosum was observed in 15 patients (94%), which can seemingly be the first sign of developing atrophy in the future. In the SPECT examination, all of the individuals exhibited asymmetric hypoperfusion in the frontoparietal lobes. The left frontal area was linked to reduced planning time without affecting strategy implementation. The right frontal lobe was engaged in making adjustments to the previously established plan. Therefore, it has an influence on visuospatial working memory, which is necessary during execution, except in the initial planning phase [61]. Cerebellar hemisphere hypoperfusion was identified in 10 patients (63%), occurring on the side contralateral to the affected cerebral cortex [62]. Cerebral blood flow asymmetry in this group of subjects was also confirmed, with greater involvement of the frontoparietal cortex and subcortical structures [63]. Additionally, hypoperfusion in the basal ganglia, particularly the putamina, was observed in 11 patients (69%), and hypoperfusion in the thalamus was noted in 14 patients (88%). CBS patients exhibited the greatest absolute differences in SPECT perfusion between the right and left sides. PSP-P patients presented the greatest variation in perfusion values between brain regions: they were highest in the insular lobe and lowest in the temporal lobe. The basal ganglia in the right brain hemisphere are more involved in the retrieval of lexical items, and they act to suppress right frontal activity to keep it from interfering with word generation processes in the left hemisphere [64]. Individuals with PSP-RS and CBS who had insular and temporal lobe involvement showed a tendency to possess nearly identical absolute SPECT perfusion differences between the left and right sides [39]. In a meta-analysis of functional MRI studies on empathy conducted by Fan and colleagues, it was discovered that the right anterior insula is linked to the affective-perceptual type of empathy. In contrast, the left insula is involved in both the affective-perceptual and cognitive-evaluative types of empathy [65]. SUVRs (standardized uptake value ratios) were determined for the striatum and the caudate and putamen separately and compared among the study groups. In addition, hemispherical and caudate-putamen differences were evaluated in atypical parkinsonism cases. After consolidating various forms of atypical parkinsonism into a unified group, the study delved into individual assessments of striatal metabolism among these distinct types. Across all types, the highest SUVRs were observed in patients with MSA (striatal SUVR: 1.50 ± 0.02). In the caudate, lower SUVRs in the left hemisphere were evident for all atypical parkinsonism types, with statistical significance observed in CBD and MSA. In contrast, the putamen displayed higher SUVRs in the left hemisphere compared to the right, with statistically significant differences across all parkinsonism types. Following a similar caudate pattern, in the striatum, the SUVRs were once again lower in the left hemisphere than the right, and statistically significant differences were observed in the MSA and CBD groups. Relying exclusively on SUVRs from the caudate and putamen and aiming to predict the specific type of atypical parkinsonism, the model demonstrated statistical significance and an overall accuracy of 55.2%. It successfully predicted PSP with 80.0% accuracy, CBD with 33.3% accuracy, and MSA with 100.0% accuracy [66].

## 8. PET

The most reliable distinction in metabolic activity between APSs and PD lies in the reduced striatal glucose metabolism observed in patients with atypical parkinsonism [57]. Through the assessment of these metabolic patterns, FDG PET imaging has secured a significant position as a supportive feature in distinguishing between different subtypes of APSs [67]. To perform PET, two groups of specific tracers are in use. The first generation includes 18F-THK5317, 18F-AV-1451 (18F-flortaucipir or [18F]FDG PET), and 11C-PBB3 [13]. The second-generation PET tracers, such as PI2620 and PMPBB3 (also known as APN1607), not only bind strongly to Alzheimer’s disease brain tissue but also significantly bind to the brain tissues of those with PSP/CBD [68]. There is increased 18F-flortaucipir retention among PSP patients in the thalamus, midbrain, caudate nucleus, putamen, and globus pallidus, even when adjusting for age, with the largest effect sizes in the last structure [39]. Predominantly in the prefrontal cortices, the anterior cingulate gyrus, and the midbrain, asymmetrical or bilateral hypometabolism was observed, depending on the type of PSP [69]. Additionally, a localized region of reduced metabolic activity in the midbrain was observed in some patients with PSP using [18F]FDG PET scans [70]. In both symmetrical and asymmetrical PSP types, notable bilateral mesiofrontal hypometabolism and, to a lesser extent, dorsolateral frontal hypometabolism are observed. In contrast, hemi-PSP patients display more asymmetric thalamic and sensorimotor cortex metabolism, and the middle cingulate cortex appears more hypometabolic compared to that in symPSP patients [26]. [18F]FDG PET (F-18-AV1451 tracer) imaging in CBS reveals asymmetric reduced glucose metabolism in several areas, including the parietal lobes extending into the posterior frontal lobes, paracentral lobule, sensorimotor cortex, thalamus, basal ganglia, middle cingulate, parietal lobe, substantia nigra [30,40], and putamen [30]. The above-mentioned structures are contralateral to the clinically more affected body side. [28]. CBD subjects are characterized by asymmetrical decreases, which are more pronounced in the hemisphere opposite to the body side that is more severely affected in the cerebrum, lateral parietal and frontal regions, and thalamus, accompanied by relative bilateral enhancements in occipital regions [65]. In cases of low-level or high-variability 18F-fluorodeoxyglucose uptake for assessing metabolic activation, it is necessary to use alternative radiopharmaceuticals. 18F-dihydroxyphenylalanine (18F-DOPA) has been introduced as a marker of dopamine uptake for imaging metabolism in the brain’s basal ganglia [71]. Striatal dopaminergic deficiency is connected with APSs [72], and because of this, 18F-DOPA has found application in APS recognition. In three patients diagnosed with clinical PSP, there was a noticeable decrease in 18F-dopa in the bilateral dorsal striatum, ventral striatum, and orbitofrontal cortex, as well as in the right amygdala, when compared to normal controls. Additionally, there were no areas where 18F-DOPA was found to be increased [73]. In six individuals with CBS, distinct abnormalities with significant asymmetry were identified in the parietal cortex, the thalamus, the caudate nucleus, and the putamen of the hemisphere predominantly affected by clinical symptoms. Additionally, reduced 18F-DOPA uptake was observed to have an asymmetric pattern in both the caudate nucleus and the putamen in four CBS patients [74]. In research in which 27 individuals were examined via PET, 9 of them (33.5%) suffering from CBS exhibited abnormalities in 18F-DOPA PET, whereas semi-quantitative analysis showed putaminal asymmetry in 17 CBS patients (63%). Uniformly diminished striatal uptake was observed in both the putamen and caudate nucleus. This contrasts with the pattern seen in PSP and PD, where the reduction in striatal uptake tends to be heterogeneous. Importantly, there were no identified correlations between 18F-DOPA PET and the clinical features of the patients [75]. Amyloid PET scans do not show positive results for CBD. However, recent studies using tau PET imaging in CBD have identified tau protein accumulation in several regions, such as the supplementary motor area, midbrain, subthalamus, perirolandic area, basal ganglia, and both cerebral and cerebellar white matter. Notably, this tau accumulation is more pronounced in areas opposite to the side more severely affected by the disease [46]. Twenty-eight patients (disease duration of 5 years or less) with MSA-P exhibited a marked decrease in 18F-DOPA uptake in the bilateral striatum. Subsequent [18F]FDG PET scans revealed a notable reduction in the bilateral putamen in patients diagnosed with MSA-P [76]. In another study, MSA-P subjects showed asymmetric diffuse hypometabolism in the putamen-pallidum, with the relative sparing of the caudate nuclei, while in MSA-C patients, hypometabolism was seen in the cerebellum and brainstem. In mixed subtypes, variable hypometabolism in the basal ganglia, cerebellum, and brainstem was associated with that in frontoparietal regions [77]. The pattern associated with CBS effectively differentiates CBS from MSA but not from PSP due to the 24% overlap in spatial metabolic patterns between CBS and PSP. By assessing the extent of hemispheric asymmetry at the network level and comparing it to the PSP-related pattern, the authors succeeded in distinguishing between CBS and PSP with a specificity of 92–94% [57].

## 9. Transcranial Sonography

Transcranial sonography (TCS) is a non-invasive diagnostic method using diverse brain structures’ echogenicity to identify a range of neurodegenerative conditions, such as APSs [78]. Elevated substantia nigra (SN) echogenicity is characteristic of PD (70–90%), frequently observed in CBD (>80%), and uncommon in MSA-P (ranging from 10% to 25%), whereas lenticular nucleus (LN) hyperechogenicity is more common in the last two diseases [79]. Similar results were obtained in another study [80]. In a study including 366 individuals suffering from various PSP types, marked enlargement of the third ventricle (≥1 cm) was noticed in 71%, whereas for LN hyperechogenicity, it was 70%. SN hyperechogenicity is seen in 22% of cases [74]. Research differentiating between PSP-RS and PSP-P showed that in the first PSP type, there is almost no SN hyperechogenicity (only 1 to 26 cases), but 86% of PSP-P patients demonstrate this symptom [81]. Notably, SN hyperechogenicity is more pronounced in PD, whereas a hyperechogenic LN, as well as an enlarged third ventricle, is more common in individuals with APSs [80]. For CBD, bilateral symmetric hyperechogenic SN areas or bilateral or unilateral hyperechogenic LN regions are quite common. The SN is often enlarged (>0.25 cm^3^ vs. <0.20 cm^3^). In PSP and MSA, there is no asymmetry in the SN range, but there may be asymmetry in the LN [82]. It is worth noting that TCS is only used as an additional examination to either diagnose an APS or determine symmetry in these disorders. Furthermore, it was found that PD individuals with levodopa-induced dyskinesia (LID) had significantly higher SN echogenicity compared to those without LID, which can explain the poor levodopa responses in PSP and MSA [83]. Additionally, a recent study confirms that TCS might be a good diagnostic tool, combined with clinical and demographic characteristics, for predicting cognitive impairments in PD. Thus, TCS appears to be a promising approach to gaining a better understanding of the mechanisms and natural disease course of atypical parkinsonian syndromes. The findings are summarized in the Table 1.

## 10. Discussion

There are many hypotheses attempting to explain the cause of neurodegeneration. They include the interplay between environmental toxins and tau or α-synuclein, inflammatory factors such as oxidative stress or microglial activation [84], and vascular components, including vascular malformations or ischemic events [85]. While there is much literature on atypical parkinsonism, little is known about the symmetry in these diseases, especially when it comes to MSA’s potential symmetry. Neuroimaging studies have demonstrated a range of metabolic, structural, and functional abnormalities, leading to a better understanding of the pathophysiological mechanisms contributing to asymmetry in APSs. On the basis of MRI, a PSP degeneration mechanism was proposed based on the correlation between clinical dysfunction and white matter tract degeneration in the superior cerebellar peduncles, body of the corpus callosum, and association fibers [86]. Aside from assessing symmetry, evaluating the unpaired structures of the brain in imaging studies also seems important. The hummingbird and morning glory signs have high specificity (99.5% and 97%, respectively), but their sensitivity is relatively low (57% and 37%, respectively) [4,37,38,39]. Monitoring the progression of midbrain atrophy and the pontine-to-midbrain ratio showed high sensitivity and specificity in differentiating PSP from PD [41]. Nevertheless, there is justification for expanding the diagnostics to PSP when at least one of these signs occurs, regardless of clinical symmetry expression. Both of these signs are good indicators for distinguishing between PSP and CBS as well [87]. The asymmetry in CBS is possibly caused by a variable asymmetric reduction in striatal D2 receptor binding, which was demonstrated through SPECT examination. This finding might explain why CBS has an asymmetric disease manifestation and why this asymmetry is the most prominent among the APSs, as well as the lack of response to dopaminergic therapy, especially for motor symptoms [63]. The atrophy in MSA most often involves the pons, putamen, and cerebellum [88]. The diagnostic specificity of the HCBs and the hyperintensity of the middle cerebellar peduncle on MRI for MSA reaches 98.5%, but it can be found in other disturbances [51], so the sensitivity of this sign is relatively low (around 50%). Higher sensitivity was obtained with the “putamina rim” sign. Its sensitivity is 72%, and its specificity is around 90%. Notably, the “putamina rim” is more often asymmetrical than the HCBs [89]. It is worth noting that even in the most sensitive imaging studies performed by experienced radiologists, small changes, such as those disrupting symmetry, can remain undetected. Therefore, the risk of false results is relatively high. Presumably, the clinical symmetry in PSP and MSA, to a great extent, depends on the atrophy symmetry. The asymmetry of atrophy or hypoperfusion in specific brain structures affects different psychosocial behaviors in patients because these structures play different roles in determining behaviors such as empathy, word generation, movement planning, and others, depending on whether they are located on the left or right side.

Data can contain errors caused by incorrect diagnosis—the incorrect classification of the patient for a given disease, as is common in the case of APSs. The degree of symmetry changes over the course of PD. According to the Hoehn and Yahr scale, which illustrates the natural, typical progression of PD, the initially asymmetric manifestation of symptoms evolves into a bilateral form, which can be mistakenly interpreted as an APS [90]. Additionally, there are no specific data regarding APS symmetry evolution. Therefore, it seems that symmetry assessment is most useful at the early stage of illness. Notably, it is observed that the symmetry of abnormalities in imaging studies does not always correlate with the symmetry of symptoms in clinical examinations, which may be attributed to two factors. First, the clinical assessment of symmetry is highly subjective and based on varying criteria, leading to the potential misclassification of the disease’s level of symmetry. Second, it is possible that the clinical abnormalities might not occur simultaneously with the imaging abnormalities. Clinically, CBS is regarded (according to the latest criteria and current publications) as a highly asymmetrical disease, whereas atrophic changes on MRI are seen to become more symmetrical as the disease progresses. It is suggested that atrophy may precede clinical abnormalities, as individuals in very advanced stages of APSs are less frequently examined by experienced clinicians due to severe symptoms and severely limited mobility. It should be noted that the study groups are relatively small, which is probably due to the fact that APSs are rare in the population, and it is difficult to find a sufficiently large group of patients. Naturally, given the very limited size of the studied cohorts, drawing definitive statistical conclusions is impossible. However, summarizing the collected data from these studies reveals a clear trend in the rates of symmetry across the various types of APSs. Additionally, in some studies, APS evaluation was not based on the newest criteria dedicated to the particular entity. The symmetry evaluation is highly subjective; therefore, the results obtained may vary between different studies, especially when they are based on different diagnostic criteria. In the authors’ opinion, the cited studies do not provide sufficient information on whether the patients were receiving pharmacotherapy during their symptoms’ severity assessment and, if so, what they were taking. Many drugs may intensify parkinsonism (i.e., neuroleptics or metoclopramide), which can also potentially affect the symptoms’ symmetry. In most of the cited studies, a neuropathological evaluation had not been performed; therefore, establishing a particular diagnosis and exact statistical proportions in the context of symmetry could be flawed. However, the data indicate that there are clear differences between particular APSs in terms of the symmetry of clinical symptoms and the signs present in imaging tests.

Although, in some cases, particular disease types were not differentiated, PSP-RS appears to be the most symmetrical PSP subtype [16], and MSA typically manifests with relatively symmetrical symptoms, whereas MSA-P might express marked asymmetry, with asymmetric dystonia of the limbs and asymmetric myoclonic jerks, leading to the wrong CBS diagnosis [27]. CBS is the most asymmetrical form of APS, and the lack of symmetry is included in its diagnostic criteria. However, CBS is not always asymmetric like the criteria describe. CBS is conventionally viewed as a sporadic disorder, implying a potential genetic predisposition to the development of symmetric degeneration [26]. Because of all of the above-mentioned factors, establishing the exact and proper diagnosis distinguishing a particular APS type or IPD is not straightforward. A misdiagnosis can lead to the initiation of inappropriate drug therapy and significant health deterioration.

## 11. Conclusions

Asymmetry may initially seem like an unhelpful diagnostic tool; however, determining which side has undergone degeneration can be taken into consideration as an additional diagnostic criterion to help minimize the risk of misdiagnosis, especially in the case of an ambiguous clinical presentation in a suspected APS. It is important to assess the clinical condition after a meticulous analysis of patient symptoms, imaging tests, and current pharmacotherapy. Additionally, determining neurodegeneration symmetry may be highly significant in identifying potential functional deficits in both the motor and neuropsychological contexts. Patients with significant asymmetry in neurodegeneration, expressed through asymmetry in motor symptoms, may exhibit different neuropsychological symptoms depending on their individual brain lateralization, especially due to frequent frontal or parietal lobe atrophy in this kind of neurodegeneration. This significantly affects already-impaired cognitive functioning; additionally, it can be exacerbated throughout the progression of the disease, because it might have a critical impact on the remaining life of the individual affected by the APS. It is also worth emphasizing that there is a great need to determine whether and how the symmetry of APS symptoms changes over the course of the disease. Furthermore, due to the connection between brain lateralization and handedness, hand dominance should be routinely examined to assess potential neuropsychological damage.

## Figures and Tables

**Figure 1 jcm-13-05798-f001:**
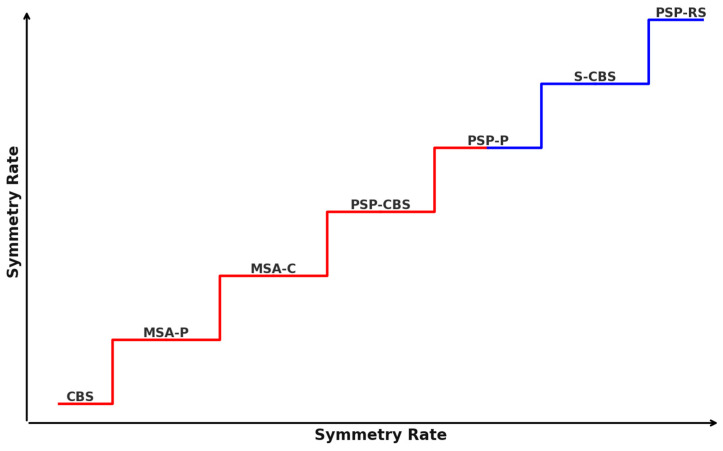
Increase in the percentage of symmetry in different types of APSs. The figure is for illustrative purposes. Red—<50% symmetry rate; Blue—>50% symmetry rate.

**Figure 2 jcm-13-05798-f002:**
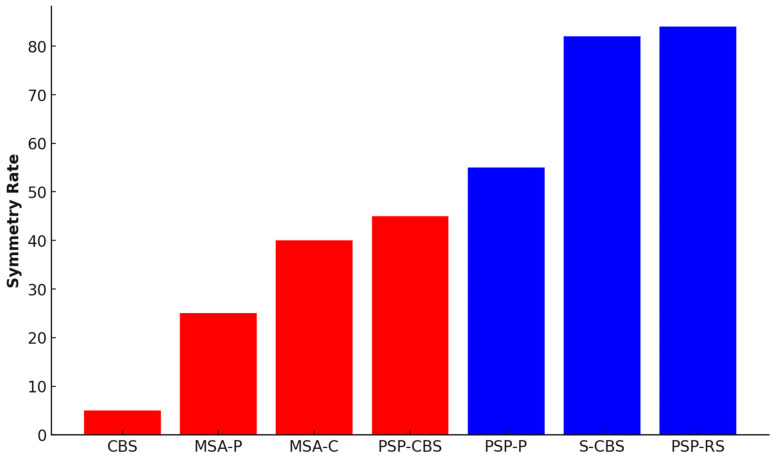
Percentage distribution of the symmetrical occurrence of clinical symptoms in the various types of APSs. CBS is presented collectively (except for S-CBS) due to the very rare occurrence of symmetrical clinical symptoms. Red—<50% symmetry rate; Blue—>50% symmetry rate.

**Table 1 jcm-13-05798-t001:** Imaging findings in APS in the context of symmetry.

Imaging	Often Occuring Disease	Finding	Morphology	Symmetry	Specificity	Sensivity
MRI	PSP-RS	Hummingbird sign/penguin sign	A midbrain tegmental atrophy without pontine atrophy	Unpaired structure	99.5%	57%
	PSP-RS	Morning glory sign	An increased lateral concavity of the midbrain tegmentum	Unpaired structure	97%	37%
	PSP	Basal ganglia, brain stem and frontal cortex atrophy		Symmetric		
	MSA	Hot cross bun sign	A linear T2-hyperintensity extending across the rostral pons	Symmetric	98.5%	50%
	MSA	Putamina rim sign	A near-total loss of neurons in the putamen	Asymmetric	90%	72%
	CBS	Basal ganglia, brain stem and frontal cortex atrophy		Asymmetric		
SPECT	PSP, MSA-P, CBS	Basal ganglia, brain stem and frontal cortex	Hypoperfusion	Asymmetric (less than in PD)		
PET	PSP	Pimple sign	A reduced metabolism in the midbrain	Unpaired structure		
	PSP	Frontal cortices, the anterior cingulate gyrus, and the midbrain	A reduced glucose metabolism	Symmetric or asymmetric		
	CBS	Disseminated: Parieto-frontal thalamus, basal ganglia, middle cingulate, parietal lobe	A reduced glucose metabolism	Asymmetric		
	MSA-P	Striatum	A reduced glucose metabolism	Symmetric or asymmetric	95%	
	MSA-C	Cerebellum and brainstem	A reduced glucose metabolism	Asymmetric	95%	
	Undifferentiated (mixed) MSA	Cerebellum, brainstem, stratium, fronto-parietal regions	A reduced glucose metabolism	Asymmetric	95%	
TCS	PSP and MSA	SN	Hyperechogenicity	Symmetric	PSP (86%) PSP-RS (3.8%)	
	PSP and MSA	LN	Hyperechogenicity	Asymmetric	PSP—70%	
	CBD	SN	Hyperechogenicity	Symmetric		
	CBD	LN	Hyperechogenicity	Symmetric or asymmetric		

## Data Availability

Data available in a publicly accessible repository.

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
