# Peer review of "Asymmetry in Atypical Parkinsonian Syndromes—A Review"

_jcm, 2024, doi:10.3390/jcm13195798_

Round 1

Reviewer 1 Report

Comments and Suggestions for Authors

The provided manuscript offers a comprehensive overview of the role of symmetry in differentiating atypical Parkinsonian syndromes (APS) from Parkinson's disease (PD). The authors have evidence from various neuroimaging techniques, highlighting the distinct patterns of asymmetry observed in different APS subtypes. The manuscript is well-structured and provides valuable insights into the diagnostic challenges associated with APS.

Here are my comments:

1. The manuscript would benefit from thorough proofreading to correct minor grammatical errors and typos.

2. The authors should ensure consistency in terminology throughout the manuscript.

3. The language could be simplified to improve clarity and readability. The text is hard to read; it is full text without pictures or tables and has more abbreviations.

4. Inclusion of a Table or Figure. A table or figure summarizing the key findings would greatly benefit the manuscript.

5. The field of neuroimaging is rapidly evolving, and it would be beneficial to include more recent literature, particularly regarding the use of advanced imaging techniques.

6. A comprehensive list of abbreviations can improve clarity and readability.

7. Please provide information about the difference between blood tests and biochemical analyses in this type of patient. 

The manuscript presents a valuable contribution to the field of APS research. However, it could be improved by including more recent literature, a discussion of limitations, and clarity and flow, which would make the manuscript even more informative and engaging for readers.

Comments on the Quality of English Language

The references should be formatted according to the journal's guidelines. The manuscript would benefit from thorough proofreading to correct minor grammatical errors and typos. The language could be simplified to improve clarity and readability. The text is hard to read; it is full without pictures or tables and has more abbreviations.

Reviewer 2 Report

Comments and Suggestions for Authors

The abstract gives a brief overview of the study and its importance but would benefit from more details about the specific questions, methods, and significance of the findings to help readers understand the research better. The introduction is crucial but could be improved by adding a thorough literature review that summarizes current knowledge on the topic, identifies gaps, and clearly states the study's objectives, making the case for its necessity. The chapters on various imaging tests are detailed and well-organized, providing a solid basis for understanding the topic. The chapter on Transcranial Sonography, is key to the study, and should be expanded with more information about techniques, applications, and recent advancements for a fuller understanding.. A section discussing the study's limitations is important for context, addressing potential biases and methodological issues, and suggesting areas for future research to give a balanced perspective. The conclusions effectively summarize the key findings and link them to the study's goals, offering valuable insights for both practitioners and researchers. The reference list is adequat.

Reviewer 3 Report

Comments and Suggestions for Authors

Firstly, I would like to thank the authors for their interesting and relevant work.

I have a few suggestions to the manuscript bellow:

1)    Please include study design type within the title.

2)    Provide a structured abstract, explain main study hypotheses in a summarized manner in the introduction/background, methods, results and discussion.

3)    Please further explain your study rationale and why you decided to study symmetry specifically

4)    We suggest using English editing services to improve readability of your article

5)    We suggest further elaborating the introduction and including more relevant studies to provide background on the presentation of atypical movement disorders, including more info on diagnostic criteria for example and more epidemiological data.

6)    We suggest further elaborating the research hypotheses in the last paragraph of the introduction.

7)    Please avoid generalizations and specify as much as possible. For example, further characterize “psychological disturbances” and “parkinsonism” on page 2

8)    Please include a detailed methods section explaining how you conducted your analyses

9)    Please include tables with the pertinent clinical and epidemiological findings of your study and compare them in a analytical way in the discussion

10) Please include a table with imaging findings comparing APS and symmetry

11) Please further elaborate discussion session including a critical analysis of the literature and summarizing new and relevant findings

Comments on the Quality of English Language

Moderate English editing required

Reviewer 4 Report

Comments and Suggestions for Authors

The article “Asymmetry in atypical parkinsonism syndromes” by Patryk Chunowski, Natalia Madetko-Alster, and Piotr Alster discusses the significance of asymmetry in the clinical manifestation and neuroimaging of atypical parkinsonism syndromes (APS), including progressive supranuclear palsy (PSP), multiple system atrophy (MSA), corticobasal syndrome (CBS), and dementia with Lewy bodies (DLB). I recommend a minor revision.

Comments:

Introduction: The introduction is well-written and frames the clinical challenge of diagnosing APS. However, a stronger connection between the significance of asymmetry in motor symptoms and its diagnostic implications would clarify the paper’s intent (considering that readers might have a strong basic science understanding but limited patient interaction).

Results: The discussion of asymmetry in CBS and its role in diagnosis is well stated, especially in motor symptoms like dystonia and apraxia.

Neuroimaging Section: The review of MRI and SPECT findings is detailed, but a deeper discussion highlighting the sensitivity and specificity of imaging findings related to asymmetry would give a better sense of their clinical value.

Discussion (Lines 421-550): The authors make a great case for the diagnostic utility of asymmetry, especially in distinguishing APS from PD. However, the discussion would benefit from a more critical evaluation of the limitations in cases where asymmetry evolves.

General Feedback:

  • The title is appropriate and accurate.
  • The abstract provides a good overview but would benefit from more specifics about the diagnostic utility of asymmetry.
  • The methods are sound but could benefit from a more detailed explanation of the imaging modalities used and their relevance to specific APS subtypes.
  • The conclusion could be strengthened by emphasizing the need for further research into how asymmetry evolves in APS and its impact on long-term outcomes.

Overall: This article provides a thorough examination of the role of asymmetry in diagnosing APS. 

Comments on the Quality of English Language

Needs Minor editing

Round 2

Reviewer 1 Report

Comments and Suggestions for Authors

Dear authors, thank you for your revision of the manuscript.

Only one comment: Please make Table 1 smaller - it is impossible to have three pages.

Thank you for your work.

Reviewer 3 Report

Comments and Suggestions for Authors

The article improved after revisions.
